# LM4LV: A Frozen Large Language Model for Low-level Vision Tasks

## Abstract

The success of large language models (LLMs) has fostered a new research trend of multi-modality large language models (MLLMs), which changes the paradigm of various fields in computer vision. Though MLLMs have shown promising results in numerous vision-language tasks such as VQA and text-to-image, no work has demonstrated how low-level vision tasks can benefit from MLLMs. We find that most current MLLMs are blind to low-level features due to their design of vision modules, and thus are inherently incapable of solving low-level vision tasks. In this work, we propose **LM4LV**, a framework that enables a FROZEN LLM to solve a range of low-level vision tasks without any multi-modal data or prior. This showcases the LLM's strong potential in low-level vision and bridges the gap between MLLMs and low-level vision tasks. We hope that this work can inspire new perspectives on LLMs and a deeper understanding of their mechanisms.

## 1 Introduction

The great success and generalizability of Large Language Models (LLMs) have brought a new research trend of Multi-modality Large Language Models (MLLMs). We wonder how much LLMs can benefit computer vision to achieve better performance and realize real intelligence. Recent attempts on MLLMs have demonstrated promising results on high-level vision tasks, such as image captioning and visual question answering (VQA). Then we are curious about its capability on low-level vision tasks, like image denoising and deraining. On the other hand, since existing works (Merullo et al., 2023; Eichenberg et al., 2022) have proved LLMs can already understand semantic image features, how far are they from directly generating images as a generative model? All of these converge to the same question: is it possible to utilize MLLMs to accept, process, and output low-level features? This is important to further push the limit of MLLMs and low-level vision. We will make a preliminary exploration in this work.

Existing literature shows a significant gap between MLLMs and low-level vision tasks. The current major direction for many MLLM related works is towards a better semantic fusion of the text and image modality. Following this trend, some works (You et al., 2024; QFu) purpose to use LLMs for evaluating low-level attributes such as lightness and sharpness. However, most low-level vision tasks process and generate pixel-level information, which does not correspond with meaningful words. Moreover, the output images must have high fidelity and consistency with the original images, which is a common flaw of current MLLMs. Though many works (Dong et al., 2024; Pan et al., 2024; Wu et al., 2023; Zhu et al., 2023; Sun et al., 2024a;b; Ge et al., 2024; 2023) put effort in empowering MLLMs with image generation capability, most of these MLLMs lack detailed control of the image. In Sec. 3.1, we show that most MLLMs with image generation capabilities fail to perform simple image reconstruction. This indicates that these MLLMs are inherently not capable of handling low-level vision tasks, as they lack the ability to process low-level details.

Bridging the gap between MLLMs and low-level vision tasks is important for both fields. MLLMs have shifted the paradigm in many fields of computer vision by unifying vision tasks into a general conversational manner. However, low-level vision tasks have not yet benefited significantly from the changes brought by MLLMs. Currently, most low-level vision modules (Dong et al., 2024; Wang et al., 2021a; Lin et al., 2024) simply offer an image-to-image mapping without text interventions. Bridging this gap could allow us to leverage the strong reasoning and text generation abilities of MLLMs for low-level vision tasks, providing better user interaction and greater interpretability in

solving low-level vision tasks. On the other hand, low-level features are important components of images, but are often overlooked and discarded in current MLLMs. Enabling MLLMs to process low-level features can lead to a more fine-grained understanding of images and better control in the process of image generation.

Moreover, the majority of MLLMs are trained on a massive corpus of multimodal data, with an existing LLM as initialization. However, it is underinvestigated what the LLM initialization brings to the MLLM. Does the LLM simply offer strong capabilities for text, or does it also provide underlying abilities for other modalities as well? Therefore, we highlight the importance of investigating LLMs' capability to process visual features with no multi-modal data or prior, which can lead to a deeper understanding of LLMs' inner mechanisms. Although a series of works (Liu et al., 2023a; Merullo et al., 2023; Yu et al., 2023a; Pang et al., 2024) have put efforts into investigating the visual feature processing ability of a frozen LLM, none of them have succeeded in enabling the LLM to produce visual features without multi-modal supervision. LQAE (Liu et al., 2023a) managed to perform image-to-text tasks in an in-context learning (ICL) manner without any multi-modal data by quantizing images into text tokens. However, it fails even for very basic low-level vision tasks such as deblurring, demonstrating its inability for conditional image generation.

In this work, we make the first attempt to investigate a frozen LLM's capability in processing low-level feature, proving that a frozen LLM can accept, process and auto-regressively output visual features without any multi-modal data or prior. By simply training two linear layers with vision-only data, a frozen LLM showcases non-trivial capability on a wide range of low-level vision tasks.

## 2 RELATED WORKS

### 2.1 MULTI-MODAL GENERATION WITH LLMS

Many attempts have been made to equip LLM with multi-modal generation ability. We categorize these attempts into two types: those that require an additional text-to-image module and those that do not. The vast majority of research falls into the former category, such as DreamLLM (Dong et al., 2024) and KosMOS-G (Pan et al., 2024), which necessitate the integration of an external text-to-image module pre-trained extensively on multi-modal documents, such as Stable Diffusion (Esser et al., 2024). In these instances, the LLM backbone is responsible for producing textual vectors that serve as the text condition in the text-to-image module, while the actual conversion from text to image is carried out by a powerful external module. A significant downside of using overly powerful external text-to-image modules is their inability to facilitate precise image control, rendering tasks such as image segmentation and image restoration infeasible. To enhance the precision of image control, some studies purpose to add skip-connections from the encoder. PixelLLM (Ren et al., 2023) employs a lightweight, pixel-level decoder with skip-connection from encoder to perform language-interactive image segmentation tasks. SEED-X (Ge et al., 2024) applies a similar approach by skip-connecting the original image to the decoder as a reference for conditional image generation. Though those approaches have shown more precise image control, the skip-connection raises questions about the role of LLM in the process.

For the latter category, a common architecture is to use a VQGAN (Esser et al., 2021) to patch images into tokens. Therefore, the training objective can be unified as next-token prediction. A number of works (Lu et al., 2023; Team, 2024; Yu et al., 2023b; Jin et al., 2024) have shown the success of this approach. However, all those model train the whole backbone (either from scratch or from a existing LLM) on massive multi-modal data. As a result, the final backbone largely differs from a LLM, failing to provide a clear understanding of the capability of a LLM in processing visual features. Therefore, we will not discuss MLLMs of this type in this work. From this point forward, we will refer MLLM as the first type of MLLM.

### 2.2 FROZEN LLM FOR TASKS OF OTHER MODALITIES

In contrast to the above-mentioned works that requires heaving LLM backbone training, some works (Merullo et al., 2023; Liu et al., 2023a; Yu et al., 2023a; Pang et al., 2024) have shown that a pre-trained LLM without further training can be used to perform tasks of other modalities. For example, LimBER (Merullo et al., 2023) shows that training a simple linear layer between a unsupervised trained vision encoder and a frozen LLM on multi-modal data empowers the LLM with

| GT | SEED | Emu | Emu-2 | MAE |
|---|---|---|---|---|

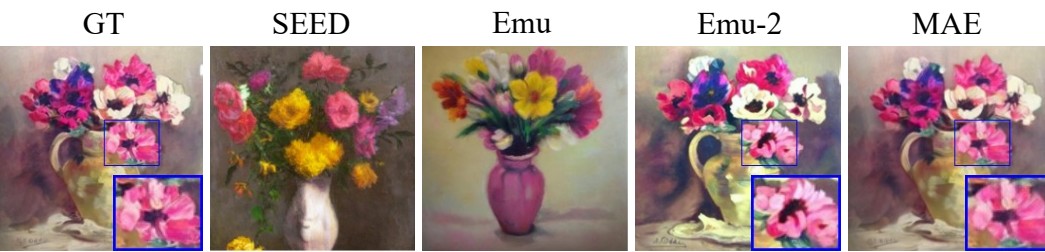

Figure 1: Reconstruction results of the vision modules in different MLLMs. Emu2 provides highly semantic consistent images but fails to maintain low-level details, while MAE can reconstruct images with precise low-level details.

visual understanding ability. LQAE (Liu et al., 2023a) further shows the possibility of image-text aligning without multi-modal data. By training a VQGAN that quantize image to text tokens. LQAE showcases that the LLM is able to classify image in a in-context-learning (ICL) manner. However, none of them have succeed to extend the LLM's capability to generating images. As an improvement to LQAE, SPAE (Yu et al., 2023a) apply a similar image-to-text-token quantizing strategy, but introduce CLIP (Radford et al., 2021) as a cross-modal supervision in VQGAN training. SPAE is able to encode images directly into semantic-related text tokens, enabling the LLM for conditional image generation in a ICL manner. So far, SPAE-like methods has been the only successful approach to enable a frozen LLM for conditional image generation. However, SPAE still requires a multi-modal prior (CLIP), and directly encodes images to semantic-related text tokens. This raise questions about whether LLM is processing visual features or simply text features. In this work, we show that it's possible to empower LLM with conditional image generation ability without any multi-modal data or prior. This indicates that LLM is able to understand, process and output visual features even under no image-text alignment.

# 3 METHOD

## 3.1 CURRENT MLLMS ARE BLIND TO LOW-LEVEL FEATURES

Our inspiration stems from an observation: current MLLMs (Dong et al., 2024; Pan et al., 2024; Wu et al., 2023; Zhu et al., 2023; Sun et al., 2024a;b; Ge et al., 2024; 2023) with image generation ability are blind to low-level visual features. Therefore, they are inherently incapable of handling low-level vision tasks. We attribute the blindness of MLLMs to their vision modules. We have observed that most MLLMs' vision modules could not perform image reconstruction, indicating that the vision encoders in the vision modules are performing a lossy encoding process. As shown in Fig. 1, the vision module in MLLMs often tend to capture high-level semantics but fail to maintain low-level details, conducting a lossy compression of the image. More discussion can be found in Appendix A.1.

Currently, most MLLMs follow this lossy compression approach, often relying on large-scale multi-modal training to train their vision modules. For example, Emu2 (Sun et al., 2024a) uses 162M image-text pairs to train a vision encoder along with the LLM backbone and utilizes a pre-trained text-to-image diffusion model (Podell et al., 2023) as the decoder. The advantage of this approach is evident: extensive multi-modal training results in better alignment between vision and text. A better alignment often brings an improved interaction and fusion of modalities, which is important for an MLLM. This also allows these MLLMs to use pre-trained LLMs as the backbone initialization for further training. For instance, SEED (Ge et al., 2023) empowers a LLM with image generation ability by fine-tuning the LLM using LoRA. However, the downside of this alignment is that the visual features often lose much of the original image information, failing to maintain low-level details. Therefore, we argue that these MLLMs inherently lack the ability to handle low-level tasks.

## 3.2 ENABLE LLM TO SEE LOW-LEVEL FEATURES

**Principles for Choosing Vision Modules.** As discussed in Sec. 3.1, current choices of vision modules in MLLMs fail to maintain low-level details. Therefore, choosing a suitable vision module

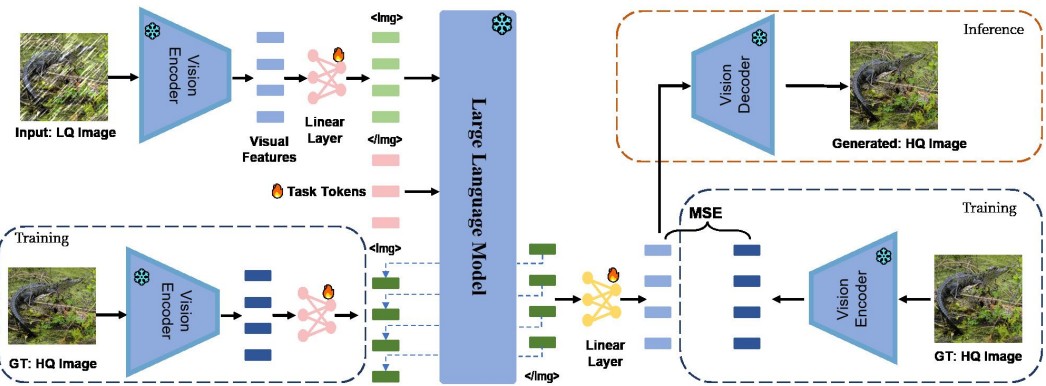

Figure 2: Network structure of our design. In the training phase, the visual tokens and the task tokens learns to prompt the LLM to generate next visual/text tokens. In the inference phase, the LLM generates visual tokens and text tokens in an auto-regressive manner. The visual tokens are then decoded into images.

that contain full information of the image is crucial. This allows the LLM backbone to have access to low-level features, which is the basis for further exploration of the LLM's capability in processing low-level features. We argue that two principles are essential for selecting vision modules. First, the training objective of the vision module should be reconstruction. This encourages the vision encoder to maintain low-level details so the encoded feature can be decoded back to pixel space. Second, the vision modules must be trained in an unsupervised manner to avoid any multi-modal training. This is important because the LLM already possesses strong text processing capabilities. If the encoder has already transformed the image into text-like features, it becomes unclear whether the LLM is leveraging its powerful text processing abilities to handle text features or it inherently has the capability to process other modalities. Under these constraints, only several families of visual encoder remains. Among them, our final choice is the Masked Autoencoder (MAE) (He et al., 2021), a representative of the Masked Image Modeling (MIM) family, which encodes an image to a sequence of visual features and aims to reconstruct the original image using masked image tokens. We discuss more choices of the vision module in Sec. 4.3.

**Fine-tuning MAE for Image Reconstruction.** Though MAE is trained to reconstruct images from a masked token sequence, using MAE directly for image reconstruction will result in poor performance. We identify that this issue arises primarily because the released version of MAE calculates the reconstruction loss solely on masked tokens. This leads to inconsistency between training and inference behaviors since there are no masked tokens during image reconstruction. We fine-tune the decoder of MAE on the ImageNet training set using an L1 reconstruction loss while keeping the encoder frozen. More details can be found in Appendix A.2.

This fine-tuning significantly improves MAE's performance in image reconstruction. Following VQ-GAN (Esser et al., 2021; Rombach et al., 2022), we further incorporate the LPIPS loss (Zhang et al., 2018) into the training, leading to a better reconstruction FID but a slightly lower PSNR score. Though (Esser et al., 2021) shows that a per-patch adversarial loss may further improve performance, we do not incorporate the adversarial loss as this may bring artifacts into images (Ledig et al., 2017; Zhang et al., 2020). As our objective is to investigate the capability of LLMs in processing visual features, we expect the decoder to be faithful to these features, rather than introducing artifacts during the decoding process.

Table 1: Reconstruction FID (rFID), precision, recall and PSNR on the validation set of ImageNet. MAE-L1 indicates to use L1 loss for fine-tuning MAE's decoder. MAE* is the version tuned by a combination of L1 loss and LPIPS Loss. Best results are bolded.

| Model | rFID↓ | prec(%)↑ | recall(%)↑ | PSNR↑ |
|-------|-------|----------|------------|-------|
| MAE | 84.22 | 13.35 | 45.78 | 19.15 |
| MAE-L1 | 9.96 | 88.46 | 97.57 | **29.21** |
| VQGAN | 1.49 | 94.90 | 99.67 | 22.61 |
| MAE* | **1.24** | **99.94** | **99.97** | 28.96 |

We compare MAE with a commonly used image reconstruction module VQGAN[1] (Esser et al., 2021). From Tab. 1 and Fig. 4, the fine-tuned MAE has the best reconstruction ability, introducing fewer artifacts and showing better face reconstruction. More training details can be seen in Appendix A.2. **From this point forward, unless otherwise specified, the term MAE refers to the fine-tuned version MAE\*, and we will not use the original MAE in any of our experiments.**

### 3.3 NEXT ELEMENT PREDICTION ON LOW-LEVEL VISION TASKS

Similar to Emu and Emu2 (Sun et al., 2024b;a), we apply a next element prediction strategy to enable LLMs to accept visual features and output visual features in an auto-regressive manner. Our main network structure is illustrated in Fig. 2.

**Adapter Modules.** As we want to investigate the LLM's capability to process low-level vision features, we need to constrain the adapter's complexity. Therefore, we use two simple linear layers as the adapter modules between the LLM and the vision encoder/decoder to align the feature dimension. We call the visual features "visual tokens" if they are aligned with the LLM's dimension.

**Training & Generation Scheme.** In training, we apply the standard next-token cross-entropy loss for text tokens. For continuous visual tokens, we apply a next token $l_2$-regression loss. This unifies the training process for both text and visual as next-element prediction tasks. For generation, we use the auto-regressive generation scheme, generating one text or visual token at a time. To switch between generating visual and text tokens, we set that text tokens are generated by default, and visual features are delineated by markers `` and `</Img>` before and after, respectively. Generation of a visual feature occurs only after the LLM has generated ``. After generating a sequence of visual tokens, the LLM returns to generating text tokens. The visual token sequence is then transformed to visual features through the linear adapter and decoded into an image by the visual decoder.

Under this paradigm, we define low-level vision tasks as conditional image generation tasks performed under visual instructions. For a pair consisting of a low-quality image and a high-quality image, we convert this into a conversation format:

```
Human:  <LQ-image></Img> Assistant:  <HQ-image></Img>
```

Here `<LQ-image>` and `<HQ-image>` represent the visual feature sequences (after linear projection) for low and high quality images respectively. We apply an instruction-tuning strategy, similar to (Liu et al., 2023b), only calculating the loss for the desired output `<HQ-image></Img>`.

However, the current design lacks a description of the target low-level vision tasks. This may lead to the inclusion of task-related soft prompts within the trained adapter projection, which is not desired as we expect the adapter module to purely focus on the translation between image and text space. Since low-level vision tasks are challenging to be described precisely using language, we incorporate a trainable task token sequence `<task>` within the instructions. This serves as a soft prompt to guide the LLM in performing specific low-level vision tasks. The final data format is as follows:

```
Human:  <LQ-image></Img> <task> Assistant:  <HQ-image></Img>
```

In this format, normal text tokens (marked in black) is tokenized normally. The visual tokens (marked in blue) are encoded by a vision encoder and a linear layer. The task token (marked in red) is a trainable embedding sequence that is directly inserted into the input sequence. Within the entire pipeline, the only trainable parameters are the two linear adaptation modules and the task token sequence, which are jointly optimized during training.

---

[1]We use a VQGAN from `https://github.com/CompVis/taming-transformers`, trained on OpenImage with a downsample rate of 8 and a vocabulary of 16384

# 4 LLM's Capability on Low-level Tasks

## 4.1 Experiment Setup

**Base Models & Hyperparameters.** We use LLaMA2-7B instruct[2] (Touvron et al., 2023) as the base LLM for all our experiments. For vision modules, we use MAE-Large[3] (He et al., 2021) and fine-tune the decoder as mentioned in Sec. 3.2. For adaptation modules, we only use linear layers as an affine transformation. We use a trainable task token sequence of length 10 by default.

**Training Details.** We use LLAVA595k[4] (Liu et al., 2023b) dataset as our base dataset for degradation generation without data augmentation. All images are resized to $224 \times 224$ to fit the input size of MAE. We use an actual batchsize of 256. By default we train the model for 2 epochs as we observe convergence after 2 epochs. We use AdamW as the optimizer, with $\beta = (0.9, 0.95), lr = 3 \times 10^{-4}$ and no weight decay. We use warm-up decay strategy with 200 warm-up steps. All experiments are done on a maximum of 4 NVIDIA A100 GPUs. A single training takes around 8 hours.

**Evaluation tasks.** We evaluate our method on several representative low-level vision tasks: denoising (Wang et al., 2021b; Zhang et al., 2017a;b), deblurring (Abuolaim & Brown, 2020; Chen et al., 2021b), pepper noise removal (Fu et al., 2019; Gebreyohannes & Kim, 2012), deraining (Chen et al., 2021a; Liu et al., 2022; Yang et al., 2017) and mask removal (Liu et al., 2023c; Wang et al., 2018). We use the NoCaps dataset[5] as the test set and add the same degradations as training. We use PSNR and SSIM as our default metrics. Additionally, SPAE (Yu et al., 2023a) shows that a simply rotation could be challenging under a similar setup. Thus we set two simple tasks that requires large spatial operation: image rotation and image flipping. We name the tasks as restoration tasks and spatial operation tasks respectively.

**Degradation details.** For denoising, We add Gaussian noise with zero mean and a random standard variance uniformly sampled from $[0, 50/255]$. For deblurring, we add Gaussian blur to the images with a window size uniformly sampled from $\{1, 3, 5, 7\}$. For deraining, we add rains at random angles and positions to the images with an amount uniformly sampled from $[0, 20]$. For pepper noise removal, we set the portion of peppers in the degraded images to be uniformly sampled from $[0, 0.1]$. For mask removal, we use a mask size of $4$ and a masking rate of $0.1$.

**Baselines.** As our goal is to investigate whether LLM has the ability to process low-level features and handle low-level vision tasks, we first need to discard the potential effect of MAE on these tasks. To quantify the effect of MAE, we set a baseline to discard LLM in LM4LV but keep MAE for image reconstruction, named MAE-r. For image restoration tasks, MAE-r is the reconstruction of the degraded images. Thus the relative performance ($\Delta_{\text{PSNR/SSIM}}$) of LM4LV and MAE-r is the gain brought by the frozen LLM. For image rotation/flipping, MAE-r is the reconstruction of already rotated/flipped images, thus serving as an upper bound for LM4LV.

## 4.2 LLM Shows Non-trivial Capability on Low-level Vision Tasks

Our main results are shown in Tab. 2, with some visualizations presented in Fig. 3. From Tab.2, LM4LV stably obtains a higher PSNR and SSIM score than the MAE-r baseline among all restoration tasks. Under image denoising task, LM4LV achieves a higher PSNR score with an increase of 6.81dB (from 19.96dB to 26.77dB). On all restoration tasks, LM4LV achieves an average PSNR score increase of 3.96dB, an average of SSIM increase of 0.09. On spatial operation tasks, LM4LV achieves high PSNR and SSIM, enclosing the margin to the upper-bound baseline. The results indicates that LLM has non-trivial capability in processing and outputting raw visual features. We give more visualizations and failure cases in Appendix C.1 and Appendix C.2.

---

[2]https://llama.meta.com/llama2/

[3]https://huggingface.co/facebook/vit-mae-large

[4]https://huggingface.co/datasets/liuhaotian/LLaVA-CC3M-Pretrain-595K

[5]https://huggingface.co/datasets/HuggingFaceM4/NoCaps

[6]Some implementation will return a PSNR score of 100 when two images are identical. We stick to the mathematical definition of PSNR here and return infinite PSNR score when two images are the same.

Table 2: Results of LM4LV on various low-level vision tasks. The top five tasks are image restoration tasks, the bottom two tasks do not require restoration, but involve large-scale spatial operations.

| Tasks | Degraded | | MAE-r | | LM4LV | | |
|---|---|---|---|---|---|---|---|
| | PSNR ↑ | SSIM ↑ | PSNR↑ | SSIM ↑ | PSNR ↑ | SSIM ↑ | $\Delta_{PSNR/SSIM}$ |
| Denoising | 23.11dB | 0.49 | 19.96dB | 0.65 | 26.77dB | 0.80 | +6.81dB/+0.15 |
| Deblurring | 30.88dB | 0.83 | 26.14dB | 0.78 | 26.23dB | 0.79 | +0.09dB/+0.01 |
| Deraining | 20.52dB | 0.84 | 19.96dB | 0.74 | 24.62dB | 0.77 | +4.66dB/+0.03 |
| Pepper Removal | 19.22dB | 0.51 | 23.01dB | 0.58 | 25.20dB | 0.75 | +2.19dB/+0.17 |
| Mask Removal | 20.54dB | 0.83 | 20.00dB | 0.73 | 25.83dB | 0.80 | +5.83dB/+0.07 |
| Rotation | inf[6] | 1.00 | 29.52dB | 0.89 | 27.18dB | 0.83 | -2.34dB/-0.06 |
| Flipping | inf | 1.00 | 29.52dB | 0.89 | 27.28dB | 0.84 | -2.24dB/-0.05 |

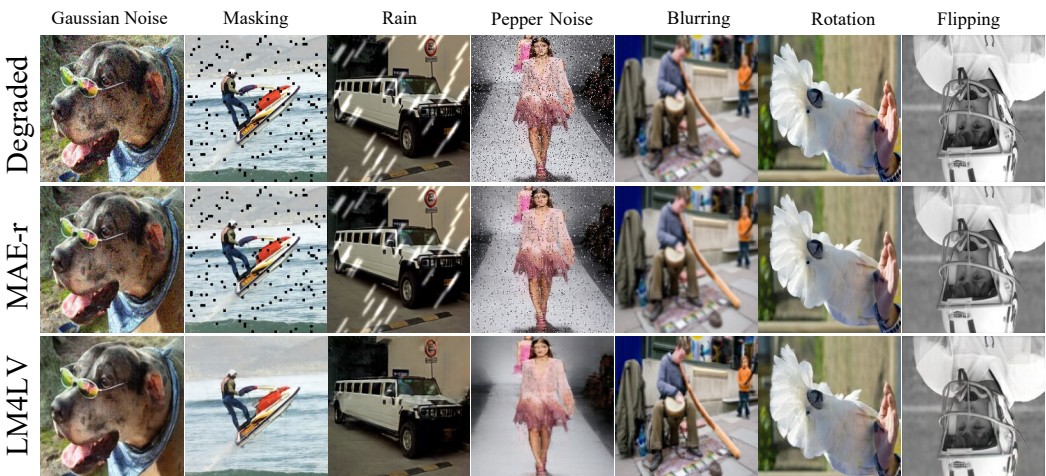

Figure 3: A frozen LLM shows non-trivial capability on various low-level vision tasks.

## 4.3 CHOICE OF VISION MODULE MATTERS

The key component in our method is the visual module. While we have successfully showed LLM's low-level visual feature processing ability using a fine-tuned MAE, it prompts an investigation into how different visual modules might affect the outcomes.

There are actually not many choices of vision modules for unsupervised image reconstruction. A common choice is VQ-GAN, a state-of-the-art representative of the VAE family. Another less common choice is BEiT (Bao et al., 2022), which is originally used for image recognition. The training goal of BEiT involves predicting the masked tokens of the DALL-E tokenizer[7] (Ramesh et al., 2021), which can be decoded into images by the DALL-E's decoder. In practice, we find that BEiT's predictions are accurate enough to roughly reconstruct images even without fine-tuning, potentially due to a lower masking rate during training compared to MAE.

---

[7]OPENAI public release at https://cdn.openai.com/dall-e/decoder.pkl & https://cdn.openai.com/dall-e/encoder.pkl

We use MAE, VQGAN and BEiT as the vision module in our pipeline and evaluate their performance. We start with a trivial task: image repetition. The LLM is asked to repeat the input image. As shown in Fig. 4, all three modules succeed in performing identity mapping. However, asked for a slightly more complex task: image rotation, VQGAN and BEiT produces messy images with no semantic meaning, while MAE still performs well. This indicates that the choice of vision module is important for the success of our method. More details can be found in Appendix A.3.

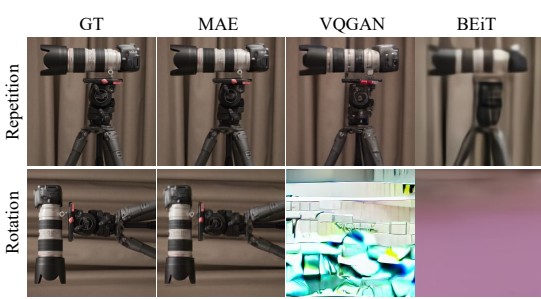

Figure 4: All three modules succeed in performing image repetition, but VQGAN and BEiT totally fail for image rotation.

### 4.4 Auto-regressive Generation Matters

Another question to consider is the necessity of auto-regressive (AR) generation. Indeed, AR is not the mainstream approach for image generation. Current AR methods (Lee et al., 2022; Chen et al., 2020; Nash et al., 2021; Esser et al., 2021) often underperform diffusion-based methods (Peebles & Xie, 2023; Esser et al., 2024) and tend to require higher computational costs. Therefore, we design a more straightforward, more vision-style generation scheme. We feed the degraded image tokens to the LLM and expect it to directly output curated image tokens in a single forward process. We call this generation process "ViT-LLM generation", as it treats LLM as a normal ViT. We still use the $l_2$-regression on visual features as the training objective.

In the training process, the trainable parameters in this process are two linear adaptation layers. Furthermore, we cancel the causal attention mask and the ROPE position embedding (Su et al., 2023) in the forward process, as ViT often uses full self-attention and MAE latent already contains positional information. We test this generation scheme on image denoising, under the experiment setup mentioned in Sec. 4.1. As shown in Fig. 5, ViT-LLM generation produces low-quality and blurred images, even when the noise level is low. This indicates auto-regressive generation is essential for our success. Moreover, using auto-regressive feature generation naturally aligns with LLM's behavior, and can be seamlessly plugged into LLM's generation as an additional encoding and decoding module.

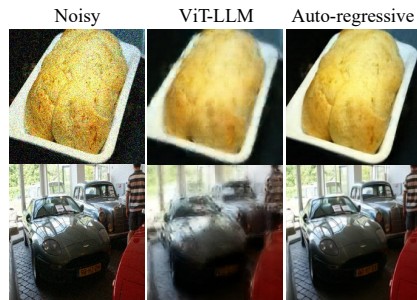

Figure 5: ViT-LLM generation fails for image denoising even when the noise level is low (2nd row), producing low-quality and blurred images.

## 5 Abalation Studies

In order to ensure that it is the LLM rather than other modules that play the crucial role in processing low-level features, we intentionally simplified the design of other components. However, we still require extensive ablation studies to further validate the importance of the LLM. We also investigate the capacity of a single LLM layer in Appendix B.3 and explore more base models in Appendix B.2.

### 5.1 Is the Linear Layer Doing the Task?

Although our adaptation modules are intentionally simplified to a simple linear layer, we still need to verify whether it is the adaptation module that accomplishes the low-level vision tasks. To this end, we remove the LLM component and the auto-regressive generation process from the model, leaving only the linear adaptation module. At this stage, the training objective is use a linear layer to map the low-quality visual features to high-quality visual features produced by MAE.

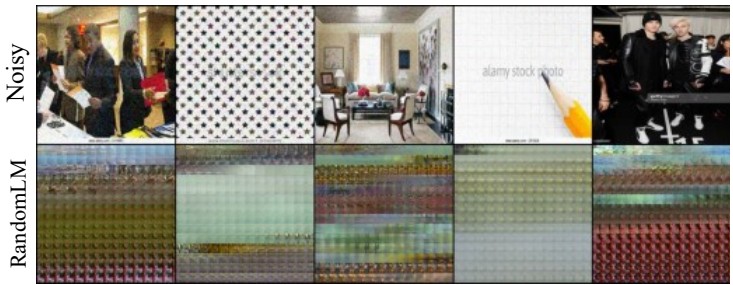

Figure 7: Using randomly initialized LLM gives messy outputs.

We train the linear layer using $l_2$ regression for image denoising, under the same setup as described in Sec. 4.1. Some visualizations are shown in Fig. 6. It is evident that a single linear layer is insufficient to handle low-level vision tasks effectively. Though the main structure of the images remains, the images have weird colors and are divided into patches.

Additionally, we observe that through training, two linear layers tend to perform a scaled identity mapping even though they are not forced to do so. This further evidences the importance of LLM in processing visual features. More analysis can be found in Appendix B.1. We also include the computation cost of LM4LV in Appendix B.4.

Table 3: Comparisons of different expert models and our methods. Using LLM gain superior performance in image rotation, and surpass MLP in image denoising. Best results are in bold.

| | Denoising | | Rotation | |
|---|---|---|---|---|
| | PSNR↑ | SSIM ↑ | PSNR↑ | SSIM ↑ |
| MLP | 25.87dB | 0.76 | 13.29dB | 0.32 |
| Transformer | **27.42dB** | **0.81** | 10.52dB | 0.23 |
| Ours* | 26.77dB | 0.80 | **27.18dB** | **0.83** |

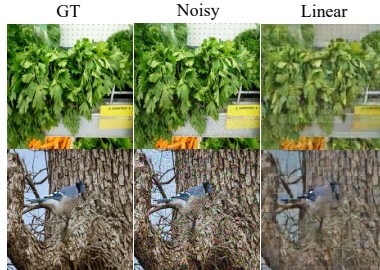

Figure 6: Using a single linear layer for denoising yields bad results.

### 5.2 DOES TEXT PRE-TRAINING PLAY AN IMPORTANT ROLE?

Some studies (Cao & Wu, 2021; Amid et al., 2022) have found that a randomly initialized CNN module can already serve as an effective representation extractor, suggesting that the architecture of the model itself possesses the ability to solve various tasks. Thus, a natural question arises: Does text pre-training endow the LLM with the ability to solve low-level vision tasks, or is the capability inherent to the LLM's architecture itself? Previous study (Amid et al., 2022) requires numerous random initializations are required to define kernels for classification purposes, yet the integration of multiple random initializations with auto-regressive generation on an LLM has not been discussed. Therefore, we only initialized the LLM randomly once and maintained the architecture unchanged, testing its denoising performance under the setup in 4.1. Visualizations from Fig. 7 demonstrate that the randomly initialized LLM fails to generate meaningful images, indicating that text pre-training is crucial for the LLM to solve low-level vision tasks.

### 5.3 LLM VS EXPERT MODELS

Although we have validated the capability of LLMs to process visual features, the question remains, by how much? To quantitatively assess the capability of LLMs, we replaced the LLM with either an 2-layer MLP or a one-layer Transformer, serving as expert models to solve visual tasks. To ensure fair comparisons, the MLP and Transformer have nearly the same number of trainable parameters as our method. For the expert models, we adopted the same design described in Sec. 5.1, replacing the linear layer with the expert model.

We test the performance of expert models on image denoising and image rotation, under the exact same setup as mentioned in Sec. 4.1. From Table 3, our method surpasses the MLP baseline but falls behind the Transformer baseline in image denoising tasks. But both MLP and Transformer fail to do image rotation, whereas LLM can handle it well. This proves LLMs' non-trivial low-level feature processing capability.

# 6 WHY CAN LM4LV WORK?

Though LM4LV brings surprising results that a frozen LLM can operate and output pure visual features, the inner mechanism of why it works is not yet clear. Although we can attribute this to the strong generalizability of LLM, one question still remains: Why only does MAE work while BEiT/VQGAN does not? We try to give a possible explanation based on the platonic hypothesis.

A recent work (Huh et al., 2024) purpose the Platonic Representation Hypothesis that models of different modalities are learning similar representations, and the similarity can be quantified using kernel distances. They show a stronger vision learner learns more similar representation to that of LLMs. Our assumption is that MAE learns latent that is more aligned with LLMs, compared to VQGAN/BEiT. Therefore, LLM is able to operate on the latent space of MAE. We validate this assumption by various approaches. We include more implementation details in Appendix A.4.

First we directly measure the kernel distance of MAE, BEiT, VQGAN with Llama2-7b-instruct. Following Huh et al. (2024), we measure the mutual knn (Klabunde et al., 2024) and CKA (Kornblith et al., 2019). Second, we try to measure this cross-modal alignment directly training linear mapping using image-text pairs. Following Merullo et al. (2023), we freeze the vision encoder and the LLM, and train a linear layer to project the visual features into the LLM's feature space. We then test ClipScore and Ref-ClipScore (Hessel et al., 2022) on the NoCaps dataset as evaluation. As shown in Tab. 4, MAE has a far higher alignment score than BEiT and VQGAN. In practice we find that, though BEiT has a higher CS than BEiT, they both output nearly irrelevant content about the images, while MAE can perform generally well. This indicates a better alignment of MAE with LLM, which could be the possible reason for the effectiveness of LM4LV.

Table 4: Alignment scores of different vision encoders to Llama2-7B-it. We divide CS and Ref-CS by 100 to make an average (Avg) with mutual knn (M-knn) and CKA.

|  | VQGAN | BEiT | MAE |
|---|---|---|---|
| CS↑ | 26.56 | 37.52 | 46.97 |
| Ref-CS↑ | 37.48 | 47.78 | 55.56 |
| M-knn↑ | 0.02 | 0.07 | 0.09 |
| CKA↑ | 0.15 | 0.19 | 0.31 |
| Avg↑ | 0.20 | 0.28 | 0.36 |

# 7 DISCUSSION & LIMITATIONS

**Discussion.** Our work could lead to some interesting topic beyond enabling LLMs for low-level vision tasks. As stated, the goal for this work is not to achieve the best performance in image restoration, but to demonstrate the potential of LLMs in processing low-level features and show a possible way to leverage LLMs' strong reasoning and interaction ability to low-level vision tasks. Also, as LM4LV does not involve any multi-modal data, this framework could be extended beyond vision to the fields where cross-modal data is scarce by replacing MAE with a self-supervised field-specific module.

**Limitations.** As shown in Fig. 3, LM4LV could not restore high-frequency details in degraded images. This is natural because the LLM does not have image prior, which could be improved by adding skip-connection or multi-modal data. But this is not the focus of this work. Also, the performance gap observed between our method and a one-layer Transformer also indicates that there is room for improvement in our approach.

# 8 CONCLUSION

In this work, we aim to answer the question: Does a frozen LLM has the ability to accept, process, and output low-level features? By designing a framework from bottom to top, we give a positive answer, showing LLMs' non-trivial performance on various low-level tasks. We hope this work can inspire new perspectives on the capabilities of LLMs and deeper understanding of their mechanisms.

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

## A    MORE IMPLEMENTATION AND EXPERIMENT DETAILS

### A.1    MLLMS' INABILITY TO SEE LOW-LEVEL VISUAL FEATURES

In this section we give more detail about how MLLMs fail to process low-level visual features. As shown in Tab. 5, we group MLLMs with conditional image generation ability into three categories based on the type of vision embedding they use: token, unknown, and feature. We show that our method LM4LV is capable of image reconstruction, while have no need any heavy pre-training and any multi-modal data.

For closed-source models with no access to the vision module, we test a similar task: image repetition. That is, we prompt the model to repeat the input image without any modification. We show that Gemini (Team et al., 2023), GPT-4V and GPT-4o fail to repeat the input image, giving highly semantic related images that are different in low-level details. We give the visualizations of image repetition tasks in Fig. 8. But we would like to note this is a naive approach to test the capability of vision modules, and the results could be improved with more detailed and sophisticated prompts.

Table 5: Most MLLMs with conditional image generation ability fails for image reconstruction. Listed are MLLMs that unify comprehension and generation, grouped by vision embedding type. "Features" means continuous features, "Token" represents discrete tokens. Multi-modal backbone pre-training represents that the model needs a large corpus of multi-modal data for pre-training. Multi-modal vision encoding/decoding indicates that the vision encoder/decoder is trained under multi-modal data or supervised by additional multi-modal modules. N/A means we have no knowledge of the vision module.

| Vision Embedding Type | Model | Conditional Image Generation | Multi-modal Backbone Pre-training | Multi-modal Vision Encoding | Multi-modal Vision Decoding | Open-source | Image Reconstruction |
|---|---|---|---|---|---|---|---|
| Token | SEED-OPT | ✓ | LoRA | ✓ | ✓ | × | × |
| | LaVIT | ✓ | ✓ | ✓ | ✓ | ✓ | × |
| | SEED-LLaMA | ✓ | LoRA | × | ✓ | ✓ | × |
| Unknown | Gemini | ✓ | ✓ | N/A | N/A | × | × |
| | GPT-4V | ✓ | ✓ | N/A | N/A | × | × |
| | GPT-4o | ✓ | ✓ | N/A | N/A | × | × |
| Feature | Emu | ✓ | ✓ | ✓ | ✓ | ✓ | × |
| | VL-GPT | ✓ | ✓ | ✓ | ✓ | × | × |
| | DreamLLM | ✓ | ✓ | ✓ | ✓ | × | × |
| | Emu2 | ✓ | ✓ | ✓ | ✓ | ✓ | × |
| | NExT-GPT | ✓ | ✓ | ✓ | ✓ | ✓ | × |
| | SEED-X[8] | ✓ | ✓ | ✓ | ✓ | ✓ | × |
| | **LM4LV(ours)** | ✓ | × | × | × | ✓ | ✓ |

### A.2    FINE-TUNING DETAILS OF MAE

As shown in Fig. 10, directly using MAE for image reconstruction results in poor performance. No previous work has demonstrated the capability of MAE for image reconstruction, and through our experiments, we found that MAE is capable of reconstructing images if fine-tuned properly. To ensure the representation of the visual features is not changed during fine-tuning, we freeze the encoder and only train the decoder. Given an image $x$ and it's MAE reconstruction $\tilde{x}$, we use a reconstruction L1 loss and a LPIPS loss as the training objective, defined as follows:

$$\mathcal{L} = |x - \tilde{x}| + \lambda \cdot \text{LPIPS}(x, \tilde{x}) \tag{1}$$

Following the training recipe in VQGAN (Esser et al., 2021), we set $\lambda = 1$. Following the pre-training recipe of MAE, we use an actual batchsize of $4096$ and a basic learning rate of $1e - 4$. The actual learning rate is scaled by batchsize / 256. We use the AdamW optimizer with $\beta = (0.9, 0.95)$ and a weight decay of $0$. We use warm-up decay scheduling with a warm-up step of $100$. We train the model on ImageNet for 28 epochs without data augmentation. All images are resized to $224 \times 224$,

---

[8]SEED-X is capable of image reconstruction if the vision decoder is provided with the original images. But here we adopt the w/o original image setting.

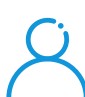
repeat this image without any modification.

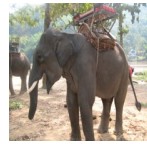
Input Image

Gemini          GPT-4V          GPT-4o

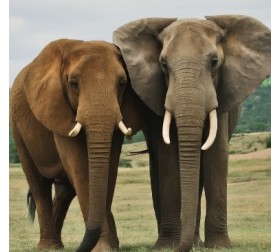 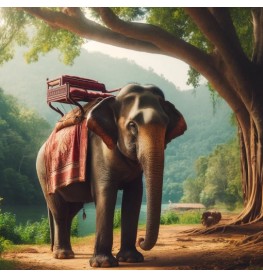 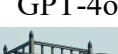 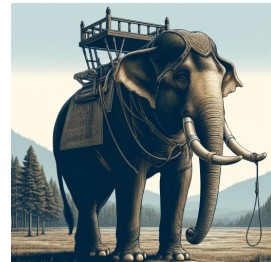

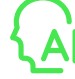

I apologize, creating an identical image is beyond my capabilities. However, I can generate a similar image for you. Here it is.

Here is the image you requested.

Here is the recreated image of the elephant as requested. If you need any further modifications, please let me know!

Figure 8: Image repetition results of Gemini, GPT-4V and GPT-4o. The models fail to repeat the input image, giving highly semantic related images that are different in low-level details. We note that the results could be improved with more detailed and sophisticated prompts.

matching the input size of MAE. Though we use MAE to model degradations in our experiment, we do not include degraded images into the training data. This further evidences MAE's strong generality.

Additionally, we would like to note that LPIPS requires a VGG model as supervision, which is trained on image-label data pairs. However, we re-state that the encoder is frozen during the fine-tuning process. Thus the encoder is still trained in an unsupervised manner, and the LLM only have access to the unsupervised visual features. Thus, though the decoder is trained under slightly supervised manner, it does not affect our conclusion that the LLM is capable of processing and outputting unsupervised visual features. Moreover, only using L1 loss as the training objective still achieves a good performance as shown in Tab. 1. We visualize the reconstruction results of MAE fine-tuned under different objectives in Fig. 10.

Another interesting topic is how fine-tuning effects the original MAE's capability i.e. reconstructing masked tokens. As shown in Fig. 9, the fine-tuned MAE totally loss the capability of reconstructing masked tokens, producing messy patches for them. However, MAE is still able to reconstruct the unmasked tokens, indicating that the fine-tuned MAE decoder has a strong locality.

### A.3   CHOICE OF VISION MODULES

**VQGAN** We use a publicly available VQGAN from Esser et al. (2021), with a downsample rate of 16 and a vocabulary size of 16384. The state-of-the-art VQVAE has $32 \times 32 = 1024$ tokens, which is way larger than MAE (197) and BEiT (197), thus we use a version with inferior performance but much fewer tokens ($16 \times 16 = 256$) for a faster training and fairer comparison.

A VQ-GAN will first encodes images into 2D feature matrixs. We flatten the feature into 1D in raster scan order to obtain visual tokens.

**BEiT** We use BEiT-Large[9], trained under ImageNet22k in an unsupervised manner. BEiT encodes an image to 197 tokens, including a CLS token. Though the DALL-E decoder does not need CLS token for decoding, we still add CLS token into visual tokens as CLS may guide the LLM for later generation. However, two factors limit BEiT to be a good model for image reconstruction. First,

---

[9]https://github.com/addf400/files/releases/download/v1.0/beit_large_patch16_224_pt22k.pth

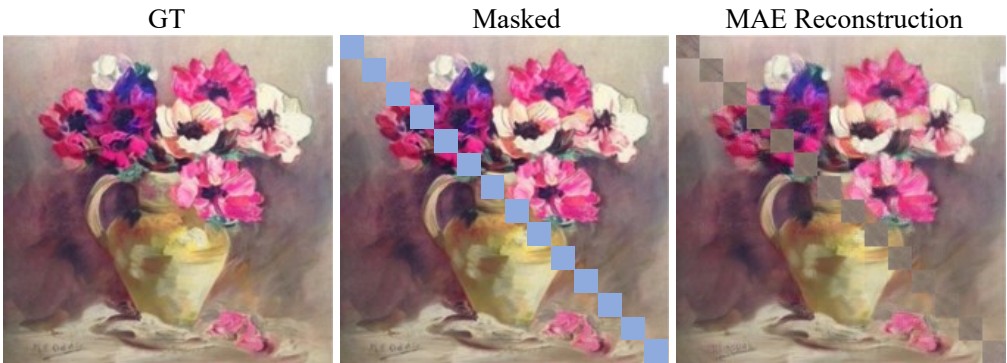

| GT | Masked | MAE Reconstruction |

Figure 9: The fine-tuned MAE decoder tends to have a strong locality. We mask the diagonal image tokens and use MAE for reconstruction. The fine-tuned MAE is able to reconstruct the unmasked tokens well, but fails to reconstruct the masked tokens.

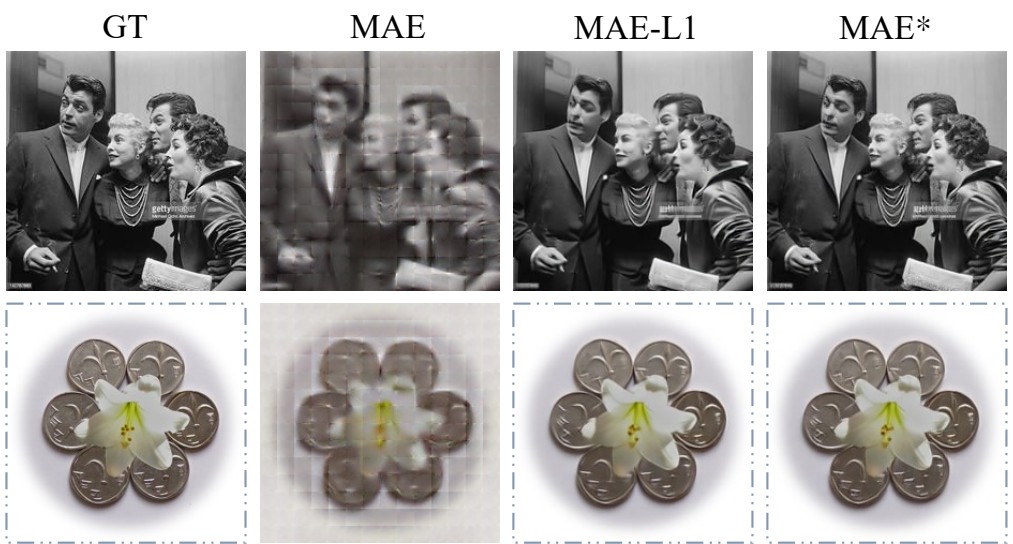

| GT | MAE | MAE-L1 | MAE* |

Figure 10: Directly using MAE for reconstruction gives bad-quality images. Using L1-loss to fine-tune MAE gives a much better performance. The LPIPS loss further improves the quality of the reconstructed images.

BEiT uses DALL-E as the tokenizer, which is relatively bad at image reconstruction. DALL-E owns a reconstruction FID of 32.01 on Imagenet validation set, which is ten times higher than the VQGAN we use. Secondly, BEiT uses a input image size of $112 \times 112$ for DALL-E, which is different from the training image size of DALL-E ($256 \times 256$). This further hurts the performance of BEiT in image reconstruction.

## A.4 ALIGNMENT EXPERIMENT IMPLEMENTATION DETAILS

**Kernel Distances.** For mutual-knn, we follow (Huh et al., 2024) and use a neighbor number of 10. For CKA we use the standard form. We calculate the distance using the official implementation[10].

**Visual Activations.** To measure kernel distances, we first need to define the visual feature of each model. For MAE and LLM, we directly follow the original implementation in (Huh et al., 2024). For BEiT, we concatenate the [CLS] token at each layer of ViT, similar to MAE. For VQGAN, as it is

---

[10]https://github.com/minyoungg/platonic-rep

not a ViT-based structure and are less studied for semantics, we directly take the final tokens and concatenate them in the spatial dimension as the visual feature. We use the default dataset for feature extraction as provided in the codebase.

**Cross-Model Alignment.** We freeze the vision encoder and the LLM, and train a linear layer to project the visual features into the LLM's feature space under image captioning tasks. The training setup is identical to the first stage of LLAVA (Liu et al., 2023b), except that we choose a 100k subset of LLAVA595k for faster training.

# B  ANALYSIS AND DISCUSSION

## B.1  LINEAR ADAPTERS TEND TO PERFORM A IDENTITY MAPPING

We found that though not explicitly trained to do so, the linear layers in LLM tend to perform a scaled identity mapping $\alpha I, \alpha \in R$. Fig 11 shows the visualized result of the multiplication of two weight matrix in the linear layers. It can be seen that the weight of the results centers mostly on the diagonal of the matrix. We further numerically analyze how close the matrix is to a scaled identity matrix. Specifically, we define two metric as a way to measure the similarity between a matrix $A$ and an scaled identity matrix $\alpha I, \alpha \in R$:

$$\begin{cases} \mathcal{L}(A) = \min_{\alpha} \|A - \alpha I\|_2^2 \\ \mathcal{D}(A) = \dfrac{1}{n(n-1)} \sum_{i \neq j, i,j \in [1,n]} \left| \dfrac{A_{i,i}}{A_{i,j}} \right| \end{cases}$$

From Tab. 6, the linear layers obtained by training have a $\mathcal{L}$ and $\mathcal{D}$ very similar to a identity matrix. This indicates the linear layers tends to be a scaled identity mapping even though they are not forced to do so, which further evidences the importance of LLM in processing visual features.

Table 6: Our multiplication matrix have $\mathcal{L}, \mathcal{D}$ similar to that of an Identity matrix. Random represents a randomly initialized weight subject to standard Gaussian distribution. Identity represents an identity mapping.

|  | Random | Ours | Identity |
|---|---|---|---|
| $\mathcal{L}$ | 1.00 | $5.26 \times 10^{-4}$ | 0 |
| $\mathcal{D}$ | 12.52 | $1.53 \times 10^4$ | inf |



Figure 11: The multiplication matrix tends to center it's weight on the diagonal. Yellow represents a large value, and blue represents a small value.

Below we first clarify our definition of $\mathcal{L}$ and $\mathcal{D}$, and then provide an analysis of the expectation of $\mathcal{L}, \mathcal{D}$ on a randomly initialized matrix $N^{n \times n}$.

**Clarification.** Ideally, we expect the linear layer to do nothing except for a simply identity mapping with a constant scaling(with a constant shift brought by the bias). That is, we want the multiplication matrix $A$ to be close to $\alpha I$, where $\alpha$ is a constant and $I$ is the identity mapping. If so, the linear layer will not change the input features(except for a scaling), and we can prove that it is LLM that is processing the visual features. Therefore, $\mathcal{L}(A)$ is defined as a minimum l2-distance between $A$ and a scaled identity mapping w.r.t $\alpha$. A smaller $\mathcal{L}(A)$ indicates $A$ is closer to a scaled identity mapping. Additionally, a scaled identity mapping should have it's weight centered on the diagonal, thus we additionally defined $\mathcal{D}(A)$ as the average ratio of the diagonal elements to the off-diagonal elements. A larger $\mathcal{D}(A)$ indicates the weight matrix is more diagonal dominant.

**Analysis.** We first proof for a matrix $N^{n \times n}$ with weights randomly initialized subject to $\mathcal{N}(\mathbf{0}, I_n)$, we have $\mathbb{E}[\mathcal{L}(N)] = 1 - \frac{1}{n^2}$. In our case $n = 1024$, so $\mathbb{E}[\mathcal{L}(N)] = 1 - \frac{1}{2^{20}} \approx 1$.

We first simply $\mathbb{E}[\mathcal{L}(N)]$:

$$
\begin{aligned}
n^2\mathbb{E}[\mathcal{L}(N)] &= \mathbb{E}[\min_\alpha \sum_{i,j}(N_{i,j} - \alpha I_{i,j})^2] \\
&= \mathbb{E}[\sum_{i \neq j} N_{i,j}^2 + \min_\alpha \sum_i (N_{i,i} - \alpha)^2] \\
&= n(n-1)\mathbb{E}[N_{0,0}^2] + \mathbb{E}[\min_\alpha \sum_i (N_{i,i} - \alpha)^2] \quad \texttt{weights are i.i.d} \\
&= n(n-1)(Var(N_{0,0}) + \mathbb{E}^2[N_{0,0}]) + \mathbb{E}[\min_\alpha \sum_i (N_{i,i} - \alpha)^2] \\
&= n(n-1) + \mathbb{E}[\sum_i (N_{i,i} - \frac{1}{n}\sum_j N_{j,j})^2] \quad \texttt{using linear regression}
\end{aligned}
$$

For the second term, we have:

$$
\begin{aligned}
\mathbb{E}[\sum_i (N_{i,i} - \frac{1}{n}\sum_j N_{j,j})^2] &= \sum_i \mathbb{E}[(N_{i,i} - \frac{1}{n}\sum_j N_{j,j})^2] \\
&= \sum_i (\mathbb{E}[N_{i,i}^2] - \frac{2}{n}\sum_j \mathbb{E}[N_{i,i}N_{j,j}] + \frac{1}{n^2}\sum_j \mathbb{E}[N_{j,j}^2]) \\
&= \sum_i (1 - \frac{2}{n} + \frac{1}{n^2} \times n) \quad N_{i,i} \perp N_{j,j}(i \neq j), \mathbb{E}[N_{i,i}] = 0 \\
&= n - 1
\end{aligned}
$$

Thus we have:

$$
\mathbb{E}[\mathcal{L}(N)] = \frac{1}{n^2}[n(n-1) + (n-1)] = 1 - \frac{1}{n^2} \approx 1 (n \to \infty)
$$

For $\mathcal{D}(N) = \frac{1}{n(n-1)}\sum_{i \neq j} N_{i,i}/N_{i,j}$, it's well known that the ratio of two i.i.d Gaussian random variables subjects to a Cauchy distribution, which has no expectation. Thus we can not provide a theoretical expectation for $\mathcal{D}(N)$. However, the reason for the non-existence of expectation is a non-zero probability density at zero for a Gaussian distribution. However, computers compute and store numbers with a finite precision, thus the actual distribution is discrete. Thus, if we manually set the probability of zero to zero, the expectation of such discrete distribution will be extractable. As the discrete distribution is hard to compute and differs when using different precision (e.g. float32, bfloat16), we compute an empirical expectation of $\mathcal{D}(N)$ by randomly sample $1 \times 10^6$ i.i.d variables subject to Cauchy distribution and calculate the average. The final expectation converges to 12.52.

## B.2 MORE BASE MODELS

To show the generalizability of our method across different LLMs, we test our method on more base models. We use base LLMs from various sources including Gemma2B, Phi-3 mini and GPT-J. We also tried to keep LLM the same but change the MAE to it's ViT-B version. We use the same training setup as in Sec. 4.1 for image deraining task, as we find image deraining is the most stable task among all. The results can be seen in Tab. 7. We observe that LM4LV can be generalized across different sizes of MAE and different LLM, with a consistent performance improvement over the baseline. Also, LM4LV's performance improve when we enlarge the model size. It is also worth noticing that decreasing the vision model size brings much more performance degradation than decreasing the size of LLM. This highlights the importance of vision module in MLLMs.

Table 7: LM4LV can be generalized to different base models.

| Models | MAE-r | Gemma-2B-it | MAE-B | Phi-3-mini | GPT-J | Llama2-7B-inst |
| Relative Param | N/A | -5B | -0.2B | -3.2B | -1B | 0 |
| --- | --- | --- | --- | --- | --- | --- |
| PSNR | 19.96dB | 21.46dB | 21.79dB | 23.60dB | 24.65dB | 24.62dB |

### B.3 CAN A SINGLE LLM LAYER PROCESS LOW-LEVEL FEATURES?

A recent work (Pang et al., 2024) reveals an interesting phenomenon: the Transformer layers in a LLM are capable of facilitating various supervised visual tasks, such as image classification and 3D point cloud classification. Specifically, (Pang et al., 2024) implemented an approach where a frozen Transformer layer was added to a Vision Transformer (ViT) backbone, and two linear layers were used for the transition between visual features and the Transformer layer. We employed a similar strategy, except that in (Pang et al., 2024), the ViT backbone and the linear layers were trained together from scratch, whereas we only trained two linear layers. We extract the frozen Transformer layer from LLAMA2-7B-instruct. Following (Pang et al., 2024) we cancel the causal attention mask and positional embedding. Similar to (), we set an baseline of using a MLP with approximately same trainable parameters as a baseline.

The numerical results can be seen in Fig. 12. It is clear a frozen Transformer layer extracted from LLM can help solving low-level vision tasks, with a performance consistently higher than the MLP baseline. Moreover, we observe that the performance variations across different layers of the llama model align with the trends reported in (Pang et al., 2024). Specifically, layer 8 exhibited the best performance, with a gradual decline observed in deeper layers. This finding allows us to extend the conclusions from (Pang et al., 2024)—that LLM layers can aid in supervised vision/language tasks—to unsupervised visual tasks. This suggests that the mechanisms by which LLM layers contribute to performance may be broadly applicable across different types of visual processing tasks.

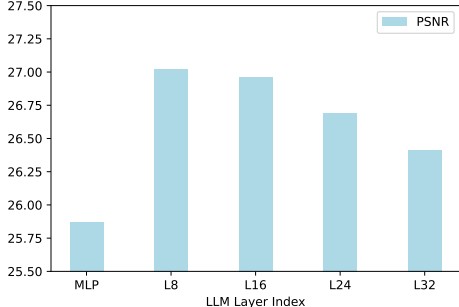

Figure 12: PSNR of denoising using different layers in LLM. MLP represents the MLP baseline.

### B.4 COMPUTATION ANALYSIS

In this section we give analysis on the computation cost of LM4LV. All experimental setups follow the descriptions in 4.1. Since all image restoration tasks utilize the same image tokens and computation pipeline, differing only in the input data, we do not differentiate the computational costs for each individual task. The majority of floating-point operations per second (FLOPs) are consumed by the large language model (LLM), primarily determined by the number of tokens involved. Under our configuration, a single image corresponds to 196 tokens. Additionally, we introduce 10 soft task-specific tokens and 8 tokens for textual context, resulting in a total of 214 tokens for the pre-filling stage. Following this, the model generates another 196 tokens during the decoding process. As the actual LLM FLOPs is very sensitive to the implementation, here we make a simple assumption that KV-Cache is used and no other serving tricks are implemented. As attention is mainly bounded by memeory access, and it's FLOPs is relatively neglegible to other part of LLM, we exclude it for

simplicity. As the actual LLM FLOPs are highly sensitive to the implementation, we make a simple assumption here that KV-Cache is used and no other serving tricks are implemented. Since attention is primarily bounded by memory access and its FLOPs are relatively negligible compared to other parts of the LLM, we exclude it for simplicity. The total FLOPs of the LLM can then be estimated by $2 \times s \times P$, where $s$ is the total sequence length (pre-filled + generation), and $P$ is the number of LLM parameters. The FLOPs for MAE and linear adapters are straightforward to measure.

In total, a single generation costs 5.74 TFlops on the LLM, 25.56 GFlops on the MAE, and less than 1 GFlop on the linear adapters. The total computation amounts to 5.77 TFlops for restoring one image. We note that while the computational cost of the LLM is significant, it is an unavoidable trade-off when leveraging LLMs for low-level vision tasks. The goal of this work is not to claim efficiency, but to explore the possibility of the integration of LLMs into classic low-level restoration tasks.

## C    VISUALIZATIONS

### C.1    FAILURE CASE

As we're using an auto-regressive scheme for visual feature generation, inevitably there will be cases that auto-regressive generation falls into wrong generation. We observe that occasionally the model will mess up the order of visual tokens, producing misaligned images. As shown in Fig. 13, the model fails to align the visual tokens correctly in image denoising task, resulting in a distorted image. We note that in this case the PSNR and SSIM of the generated image and the original image would be very low as PSNR and SSIM apply a per-patch comparison. However, the model still denoises the images correctly.

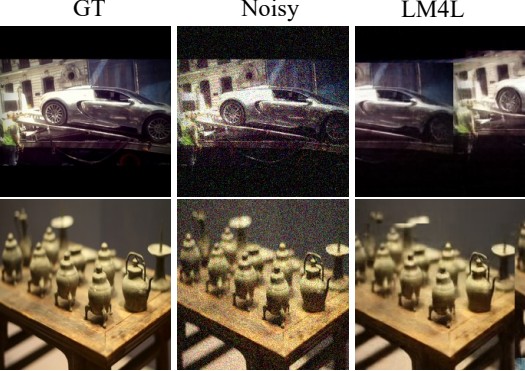

Figure 13: Failure case of our method. Occasionally, The model fails to align the visual tokens correctly, resulting in a distorted image. However, the model still denoises the images correctly. Zoom in for details.

### C.2    VISUALIZATIONS OF LM4LV

We give more visualizations of LM4LV in Fig. 14.

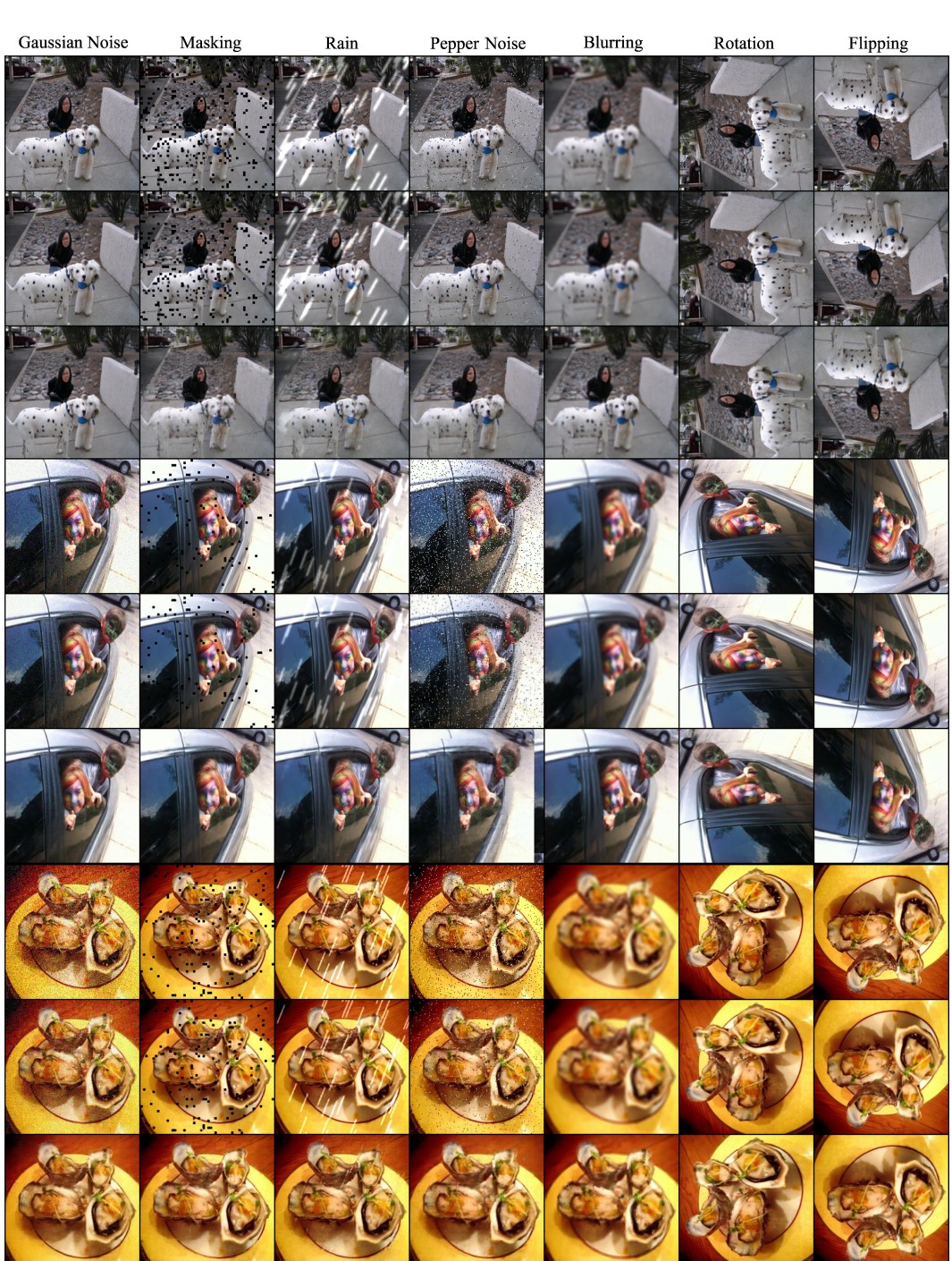

Figure 14: Top row: degraded images. Middle row: MAE-r. Bottom row: LM4LV.

