# OpenReview forum: "LM4LV: A Frozen Large Language Model for Low-level Vision Tasks"
_ICLR.cc/2025/Conference — Submitted to ICLR 2025_

### Official Review · Reviewer_Ekg6 · 2024-10-18

**Soundness:** 2
**Presentation:** 2
**Contribution:** 2
**Rating:** 5
**Confidence:** 4

**Summary:**

The paper proposes a framework called LM4LV to explore the ability of vanilla Large Language Model in handling with the low-level vision task. LM4LV utilizes the fine-tuned MAE to extract the image features, and take two linear layers as the bridge from vision to language. A paired decoder is used to transform the output tokens to images. It shows competitive quality results in denoising, deblurring, deraining and so on, but the effective in spatial operations like rotation and flipping is ordinary.

**Strengths:**

1. LM4LV explores the potential of LLM in handling with the vision data without any multi-modal data, which can provide some references.

2. Detailed appendices provide the analysis and the process of implementation. It's easy to follow it.

3. The paper is well written and easy to understand.

**Weaknesses:**

1. Contribution. The contribution is limited. LM4LV relies on the existing auto-regressive ability from the training process in LLM. It only constructs a linear layer as the bridge to align the vision semantic space and language semantic space to predict the next token.

2. Soundness. The soundness of the paper is fair. It only utilizes the LLM as a next token predictor, and not explores another way to process vision information by LLM directly.

3. The qualitative results are not fair. The comparisons with other methods should cover the other denoising, deblurring, deraining methods like MAXIM [1], Restormer [2]

4. The novelty is limited. The framework is composed with a fine-tune MAE and a LLM.

[1] Maxim: Multi-axis mlp for image processing, CVPR 22
[2] Restormer: Efficient transformer for high-resolution image restoration, CVPR 22

**Questions:**

1. In the line 54-57, paper claims that "Enabling MLLMs to process low-level features can lead to a more fine-grained understanding of images and better control in the process of image generation." It's there any fine-grained qualitative or qualitative results to prove that?

2. I can't understand the Fig.3, is there lack of labels?

3. For the auto-regressive process, the image tokens a limited in the <img> and </img>, so how to control the number of image tokens to make it same with the original image?

4. What is the order of auto-regressive, as row or column? Is there any difference when use different order to generate the image tokens?

---

> ### Author Response · Authors · 2024-11-25
> **Response to Reviewer  (1/2)**
>
> We thank the reviewer for recognizing our novel explorations and for reading the Appendix carefully. We also appreciate the acknowledgement of our writing. We will address the weaknesses and questions point by point:
>
> Weaknesses:
>
> [1] We thank the reviewer for bringing up the discussion. However, with all due respect, we disagree with the assertion that our contribution is limited. We would like to emphasize that our primary contribution is validating that LLMs possess the capability for image restoration, which is far from trivial. To ensure minimal interference from the connector between vision modules and the LLM, we deliberately use a simple linear layer. This design choice does not diminish our contribution. On the contrary, it would be less surprising if LM4LV performed well with more sophisticated modules like MLPs or Q-Former[4], as those modules may already have the capacity for image restoration, with the LLM merely acting as an identity mapping.
> In Section 5.1 and Appendix B.2, we conduct extensive experiments and analysis on the behavior of the linear layer. Our findings demonstrate that the linear layer itself cannot perform image restoration and instead tends to form an identity mapping. This further validates that it is the LLM, not the linear layer, that processes the image latents and performs the image restoration. Our contribution is also acknowledged by reviewer kc9o and reviewer ty68.
>
> [2] We agree that exploring alternative ways for LLMs to process visual information is valuable. In Section 4.4, we already investigate a ViT-like method and show that a ViT-like non-auto-regressive process fails at image denoising, resulting in messy image patches. We are more than willing to conduct additional experiments if the reviewer can propose new approaches for LLMs to process and generate visual information.
>
>
> [3] We thank the reviewer for bringing up the discussion. We would like to emphasize that the aim of this work is not to improve performance but to validate the capability of LLMs for image restoration. To ensure a rigorous validation, we deliberately constrain the trainable parameters to two simple linear layers. While this approach undoubtedly diminishes performance, it strengthens the validity of our findings.Therefore, we focus on demonstrating that LLMs achieve better performance than a "not doing anything" baseline, which in our case is MAE-r.  As mentioned by reviewer kc9o, although we do not expect LM4LV to outperform existing methods, additional comparisons with baselines could help readers better understand the position of our paper. The degradation types in Restormer do not fully align with those used in our work, resulting in a significant performance drop for Restormer. For example, in Gaussian blurring, using the ‘Single_Image_Defocus_Deblurring’ checkpoint decreases the PSNR from 30.88dB to 25.38dB. Therefore, we only present the results for image denoising, the only task that aligns with the task in Restormer, using the same data as ours and the pretrained model from Restormer. For image denoising, Restormer achieves 32.12dB, which is better than LM4LV. We hope this comparison helps the reviewer better understand the position of LM4LV. We will test additional baselines and include them in our next draft.
>
> [4] We thank the reviewer for bringing up the discussion about novelty. However, with all due respect, we disagree with the assertion that our novelty is limited. The composition of a fine-tuned MAE and LLM does not restrict our contributions.
> First, we are the first to demonstrate that MAE can be utilized for image reconstruction. While the fine-tuning process follows previous works in image reconstruction, the novel aspect lies in using MAE specifically for this purpose. Moreover, as shown in Section 4.3, other common reconstruction modules, such as VQGAN, perform poorly in comparison. In Section 6, we further explain why MAE is uniquely suitable for this task while VQGAN is not. Therefore, the inclusion of MAE is not only novel but also essential.
> Additionally, we are the first to explore the potential of allowing LLMs to directly generate visual tokens without relying on multimodal data or prior models. We also propose a training and sampling strategy that aligns traditional low-level vision tasks as a form of visual instruction tuning [3], an approach that has been largely underexplored. Therefore, we believe the novelty of our work is not limited.

---

> > ### Author Response · Authors · 2024-11-25
> > **Response to Reviewer (2/2)**
> >
> > Questions:
> >
> > [1] We thank the reviewer for the feedback and would like to clarify this claim. This claim can be broken down into two parts. First, we aim to show that current MLLMs cannot process low-level features. In Section 3.1 and Appendix A.1, we demonstrate that the vision module in many recent MLLMs fails to retain low-level image features, making them unsuitable for low-level vision tasks (even for simple image repetition). Second, we show that with a proper vision module, MLLMs can perform low-level vision tasks, even with the simple addition of two trainable linear layers, as demonstrated in LM4LV.
> >
> > [2] We apologize for any potential rendering errors. From our side, we do not observe a lack of labels. However, to address any possible issues on the reviewer's side, we have re-uploaded Fig. 3 in our revision. The first row shows the degraded images, the second row presents the results of MAE-r, and the last row displays the results of LM4LV.Columns 1-7 correspond to the tasks: ‘Gaussian Noise,’ ‘Masking,’ ‘Rain,’ ‘Pepper Noise,’ ‘Blurring,’ ‘Rotation,’ and ‘Flipping,’ respectively. It is evident that LM4LV achieves non-trivial performance across all low-level vision tasks.
> >
> > [3] The model inherently learns to generate the exact same number of tokens as the input image, as it is explicitly trained to do so during training. Specifically, the model learns to predict the token “</Img>” with very high probability after generating the same number of image tokens as the input. In practice, we do not observe any instances where the model fails to adhere to this behavior. This method is also employed by [1][2], where MLLMs are similarly used to generate image tokens in an auto-regressive manner.
> >
> > [4] We agree that it is interesting to investigate the order of tokens. For LLM, there is only one order for auto-regressive generation, as the MAE latent is already 1D (sequen length x hidden size). MAE employs a standard ViT architecture, which patchifies the image and flattens it to 1D. For MAE and most ViTs, the default patchification order is row-first. Switching to a column-first order would require modifying the patchification in MAE.
> > To the best of our knowledge, no prior works have discussed the differences between row-first and column-first patchification in MAE or the ViT structure in general. Therefore, there are no pretrained checkpoints of MAE for a column-first token order, and implementing this change would require re-training the MAE from scratch. Due to our limited computational resources, re-training an MAE from scratch is infeasible during the rebuttal period. We will make efforts to explore this and update the results in our next draft.
> >
> >
> >
> > [1] Sun, Quan, et al. "Generative pretraining in multimodality."
> >
> > [2] Sun, Quan, et al. "Generative multimodal models are in-context learners."
> >
> > [3] Liu, Haotian, et al. "Visual instruction tuning."
> >
> > [4] Li, Junnan, et al. "Blip-2: Bootstrapping language-image pre-training with frozen image encoders and large language models.

---

> > ### Comment · Reviewer_Ekg6 · 2024-11-29
> >
> > Weakness [1] Thank you for your response. My primary concern lies in understanding the source of the image restoration capability. It may be necessary to conduct a deeper analysis to determine whether this capability originates from the autoregressive generation method or the semantic space pre-trained by the LLM. A fair comparison with other autoregressive methods might also be warranted.
> >
> > Question [4] Thank you for your feedback. I understand that although MAE prioritizes row-first expansion, it utilizes a transformer for processing. This approach appears to be relatively insensitive to positional information, aside from the position encoding.

---

### Official Review · Reviewer_R61j · 2024-10-27

**Soundness:** 2
**Presentation:** 2
**Contribution:** 2
**Rating:** 3
**Confidence:** 3

**Summary:**

The paper presents LM4LV, a framework that enables a frozen large language model (LLM) to handle low-level vision tasks, such as denoising and deraining, without any multimodal data or prior. By combining a masked autoencoder (MAE) with a frozen LLM and two linear adapters, LM4LV demonstrates non-trivial capabilities in various low-level image processing tasks, thus bridging a gap between large vision-language models (LVLMs) and low-level vision.

**Strengths:**

1. The paper addresses an under-explored area in multimodal research by extending LLMs to low-level vision tasks without requiring multimodal training.

2. Comprehensive experiments cover diverse low-level tasks, including denoising, deblurring, and pepper noise removal.

**Weaknesses:**

1. The paper could benefit from a broader comparison with other non-LLM approaches, particularly in vision-language domains, to better position LM4LV’s relative advantages.

2. The motivation to use LLM for these low-level vision tasks, instead of task-specific models, are still not clear.

**Questions:**

This paper study the use of LLM for low-level vision tasks, which is under-explored. However,  the motivation to use LLM for these tasks are a bit weak, and the model struggles to preserve high-frequency details in degraded images, suggesting room for further exploration in loss functions or architectural enhancements.

---

> ### Author Response · Authors · 2024-11-25
> **Response to Reviewer and call for further discussion**
>
> We thank the reviewer for recognizing our work's focus on extending LLMs to low-level vision tasks without multimodal training and for appreciating the comprehensive experiments covering diverse tasks such as denoising, deblurring, and pepper noise removal. We’ll discuss the Weaknesses and Questions by point:
>
> Weaknesses:
>
> [1] (comparison) We want to emphasize that our work focuses on image restoration, an image-to-image task, whereas works in the vision-language domain primarily address image-to-text or text-to-image tasks. To the best of our knowledge, there are no similar approaches that employ a non-LLM text model for image restoration tasks prior to our work. We are more than willing to include additional experiments if the reviewer can propose specific baselines in the vision-language domain.
>
> [2] (motivation) We thank the reviewer for bringing up the discussion on motivation. However, with all due respect, we believe our motivation is clear. The background of this work originates from the low-level vision community's growing interest in integrating MLLMs into low-level tasks. While MLLMs have revolutionized various fields in computer vision by leveraging the strong reasoning and understanding capabilities of LLMs, the low-level vision field has yet to fully benefit from these advancements. Using traditional modules instead of LLMs can certainly achieve good performance and has been thoroughly explored over the past decade. However, such traditional modules are challenging to incorporate into the MLLM paradigm and typically adhere to a one-turn end-to-end fashion. In this work, we aim to explore a way to integrate MLLMs/LLMs into low-level vision tasks effectively. Recent works ([1], [2]) have focused on building low-level image-to-text MLLMs, but the majority of tasks in the low-level vision domain remain pixel-level image-to-image generation tasks. Our work presents the first attempt at integrating these tasks into the MLLM paradigm. Furthermore, we demonstrate that, contrary to conventional beliefs, text-only and frozen LLMs possess the capability to process and output low-level vision features. This finding expands the boundaries of what is considered possible for LLMs in low-level vision and paves the way for future research in incorporating text-only LLMs into this domain. We welcome any further discussion or suggestions from the reviewer to refine our motivation and contributions further.
>
> Questions:
>
> [1] (motivation) Please see weakness #2.
>
> [2] (high frequency details)  We thank the reviewer for pointing out the incapability of LM4LV to preserve high-frequency details. We have already discussed this limitation in the manuscript (lines 528-530): "As shown in Fig. 3, LM4LV could not restore high-frequency details in degraded images. This is natural because the LLM does not have an image prior, which could be improved by adding skip-connections or multimodal data." We encourage the reviewer to provide more detailed feedback for further improvements.
>
> [1] Q-Future/Q-Instruct: [CVPR 2024] Low-level visual instruction tuning, with a 200K dataset and a model zoo for fine-tuned checkpoints.
>
> [2] Zhiyuan You, Zheyuan Li, Jinjin Gu, Zhenfei Yin, Tianfan Xue, and Chao Dong. Depicting Beyond Scores: Advancing Image Quality Assessment through Multi-modal Language Models.

---

### Official Review · Reviewer_kc9o · 2024-10-31

**Soundness:** 3
**Presentation:** 3
**Contribution:** 3
**Rating:** 6
**Confidence:** 3

**Summary:**

The authors introduce LM4LV, a framework that enables frozen LLMs to handle low-level vision tasks (like image denoising and deraining) without requiring multi-modal data or prior training. Through comprehensive experiments, the authors demonstrate the framework's effectiveness by showing significant improvements over baselines and reveal two crucial design choices: the necessity of auto-regressive generation and the importance of vision module selection.

**Strengths:**

Pro:
1. The paper tackles an interesting research direction by investigating whether frozen LLMs can handle low-level vision tasks without multi-modal training, addressing a significant gap in current MLLM research.
2. The authors propose LM4LV, an efficient framework that achieves impressive results across multiple low-level vision tasks using only two trainable linear layers while keeping the LLM frozen.
3. The paper provides thorough empirical analysis through comprehensive ablation studies, particularly demonstrating the necessity of auto-regressive generation, the importance of choosing appropriate vision modules, and the effectiveness of each module, which helps validate the framework's design choices.

**Weaknesses:**

Cons:
1. The paper said, “Furthermore, we cancel the causal attention mask and the ROPE position embedding in the forward process, as they are not the common practice for vision modules.”. However, the ROPE and it variant 2D-ROPE are widely used in large vision transformers (e.g., EVA). This sentence needs revision, and additional experiments with position embeddings would strengthen the analysis.
2. The paper does not explore different LLM variants and sizes. While testing very large LLMs may be resource-intensive, experimenting with smaller or comparable models (like LLaMA-3.2-3B or Qwen2.5 series) would provide valuable insights into how model architecture and scale affect performance.

Suggestion:
1. The paper would benefit from including comparisons with state-of-the-art (SOTA) results from mainstream methods for each low-level vision task. The readers do not expect the method to outperform SOTA methods, while a clear comparison might be helpful for the readers to understand the position of the paper.
2. It would be valuable to investigate the effectiveness of non-autoregressive language models like BERT for low-level vision tasks. While the paper's experiments show that removing auto-regressive generation from LLMs significantly degrades performance, it alters the pre-trained model. Testing with models that are inherently non-autoregressive (like BERT) could provide clearer insights into whether the auto-regressive nature of LLMs is truly essential for these tasks or if alternative architectures might be viable with proper adaptations.

**Questions:**

NA

---

> ### Author Response · Authors · 2024-11-25
> **Response to Reviewer**
>
> We thank the reviewer for their thoughtful feedback and for recognizing the significance of our work in addressing a critical gap in MLLM research. We also appreciate the acknowledgement of our ablation on design choices. We’ll discuss the Weaknesses and Questions by point:
>
> Weaknesses:
>
> [1] (position embedding) We thank the reviewer for pointing out the misunderstanding. What we intended to convey is:
> a. Causal masking is not a standard practice in ViT.
> b. Adding positional embedding to an MAE latent is not a common practice. Since MAE already incorporates a 2D sinusoidal embedding during encoding, the latent produced by MAE inherently contains positional information. Adding an additional ROPE embedding to the MAE latent is then not a common practice.Therefore, we removed the ROPE embedding and the causal mask. But as the reviewer points out,  ROPE embedding itself is indeed a common practice in ViT. We have also updated the manuscript to provide a clearer explanation in the revised version.
>
> [2] (different LLMs) We strongly agree that exploring different model sizes is important. We have conducted such an investigation, as detailed in Appendix B.2. As the reviewer noted, experimenting with larger LLMs is resource-intensive; therefore, we utilized a diverse family of lightweight LLMs. The results are presented in Table 7 and are also summarized below.
>
> | Model           | Relative Param | PSNR     |
> |------------------|----------------|----------|
> | MAE-r (baseline)           | N/A            | 19.96dB  |
> | Gemma-2B-it     | -5B            | 21.46dB  |
> | MAE-B           | -0.2B          | 21.79dB  |
> | Phi-3 mini      | -3.2B          | 23.60dB  |
> | GPTJ            | -1B            | 24.65dB  |
> | Llama2-7B-inst  | +0             | 24.62dB  |
>
>
> Not only do we investigate LLMs of different sizes, but we also examine the effect of varying the size of MAE. Reducing the size of either MAE or LLM decreases performance; however, the performance remains above the baseline. This demonstrates that LM4LV can generalize well to other variants of LLMs and MAEs.
>
> Suggestions:
>
> [1] (baselines) We agree that comparing with traditional baselines can help readers gain a better understanding. We tested Restormer [1], a classic image restoration method designed for various degradations.  The degradation types in Restormer do not fully align with those used in our paper, which brings a significant performance drop for Restormer (for Gaussian blurring, using the ‘Single_Image_Defocus_Deblurring’ checkpoint decreases the PSNR from 30.88dB to 25.38dB. Therefore, we only present the results for image denoising, the only task that aligns with the task in Restormer, using the same data as ours and the pretrained model from Restormer. For image denoising, Restormer achieves 32.12dB, which is better than LM4LV (26.77 dB). We hope this comparison helps the reviewer better understand the position of LM4LV. We will test additional baselines and include them in our next draft.
>
> [2] (BERT) We agree that using a Masked Language Modeling model like BERT may eliminate the need for autoregressive generation. However, this work is motivated by the growing trend of applying LLMs to vision tasks, and our focus is on exploring the potential of LLMs for image restoration. But still we conducted a pilot experiment using bert-large (https://huggingface.co/google-bert/bert-large-uncased) in LM4LV. We employed the ViT-LLM method for image denoising, as described in Section 4.4, and used the same training hyperparameters as in our main experiments. We find that BERT fails to produce meaningful images, instead generating messy image patches. While further experiments could be conducted to improve performance—for example, by using a MaskGIT-like sampling method—such investigations fall outside the scope of this work. We will leave these explorations for future research.
>
> We sincerely thank the reviewer for providing many valuable suggestions. Please do not hesitate to contact us if you have any further questions or additional advice.
>
> [1] Zamir, Syed Waqas, et al. "Restormer: Efficient transformer for high-resolution image restoration."

---

### Official Review · Reviewer_HF2B · 2024-11-01

**Soundness:** 2
**Presentation:** 3
**Contribution:** 3
**Rating:** 5
**Confidence:** 4

**Summary:**

This paper proposes a novel framework, named LM4LV, which applies MLLM to low-level vision tasks. Through preliminary experiments, the authors highlight the advantages of MAE in preserving detailed visual features. Therefore, MAE is used as the visual encoder in LM4LV. Keeping the LLM and visual encoder frozen, the training of LM4LV can be conducted without multimodal data. Extensive experiments are conducted on various low-level vision tasks and different vision modules, to verify the low-level vision task capability of LLM.

**Strengths:**

+ The proposed method is simple and easy to understand.
+ This work is the first to use LLM for low-level vision tasks.
+ Some conclusions in the paper are interesting. E.g. MAE visual tokens are robust to rotation.

**Weaknesses:**

- It would be helpful if the authors could specify the number of visual tokens generated by MAE for each image. Moreover, the discussion about computation cost is missing.
- While in sec. 4.1, the authors use MAE-r model as a baseline. Since MAE-r is trained for image reconstruction only, the baseline is not very strong. To establish a stronger baseline, the authors can consider adding the linear adapters into MAE-r, and train the linear adapters for image restoration tasks. Ideally, the linear adapters should be trained separately for each image restoration task, with the same recipe used for training LM4LV.
- Lack of comparison with traditional low-level vision methods. I'm not an expert in low-level vision tasks, so a dedicated part in the related works section would be helpful. Moreover, I don't see any comparison between the proposed LM4LV framework and the previous low-level vision methods.
- The presentation contains several typos. In line 018 "we purpose LM4LV" and line 039 "purpose to use LLMs", should the word "purpose" be "propose"? In line 471 "(Amid et al., 2022) requires", the citation format can be corrected. In line 914, "ativation" might be a typo.

**Questions:**

1. Why fine-tune MAE to perform image reconstruction, instead of using MAGE or MaskGIT? MAGE and MaskGIT use the MIM training paradigm to perform image reconstruction. Will they achieve better performance than MAE?
2. In sec. 5.2, the authors claim they randomly initialize the LLM. After the random initialization, is the LLM frozen or fine-tuned? It would be interesting if the LLM is fine-tuned but still underperforms the frozen LLM counterpart.

---

> ### Author Response · Authors · 2024-11-25
> **Response to Reviewer (1/2)**
>
> We thank the reviewer for acknowledging the simplicity and clarity of our proposed method and for recognizing our work as the first to use LLMs for low-level vision tasks. We also appreciate the positive feedback on the interesting conclusions, such as the robustness of MAE visual tokens to rotation. We’ll discuss the Weaknesses and Questions by point:
>
>
> Weakness:
>
> [1] (tokens and compute): We thank the reviewer for bringing up the discussion on computation. We have included an additional section to address this in Appendix B.4 of our revised manuscript. The token amount is determined by $tokens = \frac{HW}{p^2}$, where $H$ and $W$ represent the dimensions of the image, and $p$ denotes the patch size. For our configuration, $H = W = 224$ and $p = 16$, so the token amount is 196 for a single image. The computation cost is composed of three parts: LLM computation, MAE computation, and linear layer computation, which are listed as follows:
> LLM | MAE | Linear | Total
> --- | --- | ------ | -----
>   5.74 TFlops   | 25.56 GFlops     |     <1 GFlops | 5.77 TFlops
>
> We want to note that while the computational cost of the LLM is significant, it is an unavoidable trade-off when leveraging LLMs for low-level vision tasks. The goal of this work is not to claim efficiency, but to explore the possibility of the integration of LLMs into classic low-level restoration tasks. Our contribution lies in validating the capability of LLMs on low-level vision tasks, potentially serving as a foundational step toward deeper integration of LLMs and image restoration. Consequently, the computation of the LLM is inherently involved.
>
>
> [2] (linear adapter) We strongly agree that a thorough investigation of the effect of the linear adapter in our pipeline is necessary. Tthis is what we conducted in Sec 5.1 and Appendix B.1. In Sec 5.1, we conduct the exact experiment requested by the reviewer. As shown in Fig. 6, a single linear layer fails at denoising, resulting in images with noticeable and strange patch artifacts. To further investigate the effect of the linear layer within our pipeline, we provide both numerical results and visualizations in Appendix B.1, demonstrating that the two linear adapters primarily learn an identity mapping. Therefore, it is the frozen LLM that is actually performing the image restoration.
>
> [3] (improved baselines) We agree with the reviewer that adding related work on low-level vision is beneficial. We’ll add more discussion in our next draft. Regarding improved baselines, we want to emphasize that this work is the first successful attempt to enable a frozen LLM to perform image restoration tasks. There is no prior work comparable to ours. And naturally, the performance of our method lags behind traditional methods, as we have deliberately minimized the trainable parameters and kept the LLM frozen. This natural gap is also recognized as reasonable by reviewer kc9o. The objective of this work is not to claim superior performance but to introduce a novel possibility for deeper integration of LLMs into image restoration tasks. We impose many constraints to validate that a frozen LLM has the capability to perform image restoration, and these constraints undoubtedly diminish the performance. But we agree that comparing with traditional baselines can help readers gain a better understanding. We tested Restormer [1], a classic image restoration method designed for various degradations.  The degradation types in Restormer do not fully align with those used in our paper, which brings a significant performance drop for Restormer (for Gaussian blurring, using the ‘Single_Image_Defocus_Deblurring’ checkpoint decreases the PSNR from 30.88dB to 25.38dB. Therefore, we only present the results for image denoising, the only task that aligns with the task in Restormer, using the same data as ours and the pretrained model from Restormer. For image denoising, Restormer achieves 32.12dB, which is better than LM4LV (26.77 dB). We hope this comparison helps the reviewer better understand the position of LM4LV. We will test additional baselines and include them in our next draft.
>
>
> [4] (typos)  We thank the reviewer for the thorough reading and pointing out the typos. We have already fixed them and updated a new revision.

---

> > ### Author Response · Authors · 2024-11-25
> > **Response to Reviewer (2/2)**
> >
> > Questions:
> >
> > [1]  (MAE alternatives) We thank the reviewer for bringing up this interesting possibility. In fact, MAGE was initially our preferred choice over MAE. However, we later found that both MAGE and MaskGiT reconstruct images by predicting discrete latent pixels of a VQVAE. This approach inherently limits the maximum reconstruction ability, even when prediction accuracy reaches 100%, as it depends on the reconstruction quality of the VQVAE itself.Upon investigating the VQVAE used in these methods, we found its reconstruction ability to be quite limited, as shown in Table 1. Specifically, the PSNR is only 22.61, whereas MAE achieves a significantly higher PSNR of 28.96. For this reason, we opted for MAE to ensure a better reconstruction upper bound.
> >
> >
> > [2] (LLM) The LLM is then frozen. Since the LLM is randomly initialized and not pre-trained on text, it effectively becomes a 7B vision model once fine-tuned on image restoration tasks. Naturally, such a model would perform better than an LLM trained on text. However, during the rebuttal period, we cannot implement this approach, as fine-tuning a 7B model totally exceeds our computational resource limits. Additionally, we think it is beyond the scope of this work. We will leave it for future explorations.
> >
> > We sincerely thank the reviewer for carefully reading our paper and providing many valuable suggestions. Please do not hesitate to contact us if you have any further questions or additional advice.
> >
> > [1]  Zamir, Syed Waqas, et al. "Restormer: Efficient transformer for high-resolution image restoration."

---

> ### Comment · Reviewer_HF2B · 2024-11-30
>
> I appreciate the detailed response from the authors and acknowledge the primary aim of this work: to demonstrate the feasibility of integrating LLMs into classic low-level restoration tasks. However, the experimental results seem to suggest that LLMs may not be well-suited for such tasks, as the proposed method falls short of traditional approaches in both performance and efficiency.
>
> From a conceptual standpoint, low-level restoration tasks heavily rely on detailed appearance features, and attempting to handle such features with LLMs would result in prohibitively high computational costs. This raises concerns about the practicality of LLMs in this domain. I encourage the authors to explore potential solutions to this challenge. For now, I will maintain my current score.

---

### Official Review · Reviewer_ty68 · 2024-11-04

**Soundness:** 3
**Presentation:** 3
**Contribution:** 3
**Rating:** 6
**Confidence:** 3

**Summary:**

This paper analyzes the reasons why most MLLMs cannot handle low-level features and then puts forward a simple solution. It uses MAE fine-tuning with Image reconstruction tasks to save low-level features. After that, it is then interesting to find that frozen LLM can understand low-level features, which taps into the LLM's uncanny ability to understand visual features.

**Strengths:**

(1) It proves frozen LLM can solve low-level vision task
(2) The ablation experiment is sufficient to answer that the processing of low-level information is not due to the trainable linear layer, but the text pre-training plays a role

**Weaknesses:**

vision encoder selection is relatively small, exploring more vision encoders will be more convincing

**Questions:**

None

---

> ### Author Response · Authors · 2024-11-25
> **Response to Reviewer**
>
> We thank the reviewer for recognizing our contributions, specifically demonstrating that frozen LLMs can effectively solve low-level vision tasks and showing through comprehensive ablation experiments that the ability to process low-level information stems from text pre-training rather than the trainable linear layer.
>
> Weakness:
>
> We agree with the reviewer that exploring additional vision encoders could further strengthen our study. We already conducted an experiment in Appendix B.2 to validate this. We’ll expand the discussion here. Incorporating larger vision encoders such as MAE-H presents significant computational challenges, as we need to finetune them for image reconstruction on ImageNet. Our limited computation resource cannot support us to do that during rebuttal.
> As an alternative, we conducted a reverse experiment by fine-tuning a smaller vision encoder, MAE-Base, which is substantially smaller than the MAE-Large encoder used in our primary experiments. We evaluated this smaller model on the image deraining task, where its performance dropped from 24.62 dB (achieved with MAE-Large) to 21.79 dB. Although the smaller encoder still outperformed the MAE-r baseline (19.76 dB), the results indicate a noticeable deterioration in performance as the encoder size decreases.
> This observation highlights the importance of using appropriately scaled encoders to achieve optimal results, underscoring the trade-off between computational feasibility and performance.
>
> We also conducted an additional experiment by reducing the size of the LLM. Specifically, we utilized Phi3-mini, which is 3.2B parameters smaller than the LLM used in our main experiments (LLaMA2-7B-Inst). On the same image deraining task, the performance decreased from 24.65 dB to 23.60 dB. To provide a clearer comparison, we summarize the parameter reduction and corresponding performance changes in the table below. It is evident that while the reduction in parameters for the vision encoder is much smaller than that for the LLM, the corresponding performance drop in the deraining task is significantly larger. This observation underscores the critical importance of the vision encoder in this task.
> | Model          | Parameter Reduction | Deraining Performance (dB) | Performance Drop (dB) |
> |----------------|---------------------|----------------------------|-----------------------|
> | MAE-B          | -0.2B              | 21.79                      | -2.86                |
> | Phi3-mini      | -3.2B              | 23.60                      | -1.05                |
> | LLaMA2-7B-Inst | 0         | 24.65                      | 0                   |
>
> We sincerely thank the reviewer for carefully reading our paper and providing many valuable suggestions. Please do not hesitate to contact us if you have any further questions or additional advice.

---

### Meta-Review · Area_Chair_Ls6H · 2024-12-23

**Metareview:**

The paper introduced LM4LV, a new framework for low-level vision, which uses a frozen LLM to solve low-level vision tasks. The authors use MAE as the visual encoder, combine it with LLM and keep both frozen, the training of LM4LV can be conducted without multimodal data. The experiments demonstrate the low-level vision capability of LM4LV.

This paper is interesting to me. It explored a new direction and problem, which shows that frozen LLMs with proper approach can solve low-level vision tasks to som extent.

But the concerns about insufficient ablations, low performance and efficiency, and unclear motivation/advantage do exist.

I recommend rejecting as the current state, but the decision can be bumped up.
I would suggest that the authors explore the potential advantages further. Again, I think this is a interesting exploration.

**Additional Comments On Reviewer Discussion:**

The paper received mixed ratings of 6-6-5-5-3.

The reviewers recognized the strengths such as interesting direction, under-explored area, well-written and easy to understand.
The raised weaknesses include:

- Reviewer ty68: insufficient ablations on vision encode selection.
- Reviewer HF2B: motivation, performance, efficiency, practicality of LLMs in this domain.
- Reviewer kc9o: insufficient ablations on LLM variants and sizes, insufficient comparison with state-of-the-art results for each low-level vision task.
- Reviewer R61j: a broader comparison with other non-LLM approaches, particularly in vision-language domains, the motivation to use LLM for these low-level vision tasks is still not clear.
- Reviewer Ekg6: Limited contribution and novelty, soundness is fair, unfair and insufficient comparison.

The authors provided a rebuttal for each reviewer. The rebuttal didn't change the reviewer's concerns on the current state of the paper.

---

### Decision · Program_Chairs · 2025-01-22

Reject